# p38 Mitogen-Activated Protein Kinase Inhibition of Mesenchymal Transdifferentiated Tumor Cells in Head and Neck Squamous Cell Carcinoma

**DOI:** 10.3390/biomedicines11123301

**Published:** 2023-12-13

**Authors:** Julia Federspiel, Maria do Carmo Greier, Andrea Ladányi, Jozsef Dudas

**Affiliations:** 1Department of Otorhinolaryngology and Head and Neck Surgery, Medical University of Innsbruck, Austria and University Hospital of Tyrol, 6020 Innsbruck, Austria; julia.ingruber@i-med.ac.at (J.F.); maria.greier@i-med.ac.at (M.d.C.G.); 2Department of Surgical and Molecular Pathology and the National Tumor Biology Laboratory, National Institute of Oncology, 1122 Budapest, Hungary; ladanyi.andrea@oncol.hu

**Keywords:** ralimetinib, SB201290, partial epithelial–mesenchymal transition, cisplatin, head and neck squamous cell carcinoma

## Abstract

High mortality in head and neck squamous cell carcinoma (HNSCC) is due to recurrence, metastasis, and radiochemotherapy (RCT) resistance. These phenomena are related to the tumor cell subpopulation undergoing partial epithelial to mesenchymal transition (pEMT). Repeated transforming growth factor-beta (TGF-beta-1) treatment via the p38 mitogen-activated protein kinase (p38 MAPK) signaling pathway induces pEMT in SCC-25 HNSCC cells, and activates and stabilizes the pro-EMT transcription factor Slug. We investigated the growth inhibitory, cisplatin-sensitizing, and pro-apoptotic effects of p38 MAPK inhibition in cisplatin-resistant (SCC-25) and -sensitive (UPCI-SCC090) HNSCC cell lines, using two specific p38 MAPK inhibitors, SB202190 and ralimetinib. Cell viability was measured by MTT assay; cell cycle distribution and cell death were evaluated by flow cytometry; p38 MAPK phosphorylation, Slug protein stabilization, and p38 MAPK downstream targets were investigated by Western blot. p-p38 inhibitors achieved sustained phosphorylation of p38 MAPK (Thr180/Tyr182) and inhibition of its function, which resulted in decreased phosphorylation (Thr69/71) of the downstream target pATF2 in pEMT cells. Subsequently, the p-p38 inhibition resulted in reduced Slug protein levels. In accordance, p-p38 inhibition led to sensitization of pEMT cells to cisplatin-induced cell death; moreover, p-p38 inhibitor treatment cycles significantly decreased the viability of cisplatin-surviving cells. In conclusion, clinically relevant p38 inhibitors might be effective for RCT-resistant pEMT cells in HNSCC patients.

## 1. Introduction

Head and neck squamous cell carcinoma (HNSCC) is one of the most common cancer types worldwide, with high mortality rates despite recent advances in therapeutic modalities. Poor survival can be attributed to the high frequency of recurrence and second primary tumor, to tumor metastasis and radiochemotherapy (RCT) resistance [1,2]. Hypoxic conditions, cellular stress, mitochondrial dysfunction [3], and, in particular, the reversible process of partial epithelial to mesenchymal transition (pEMT) precondition tumor cells to therapy resistance [4,5,6]. Both applied RCT and pEMT processes might activate and stabilize the pro-EMT-transcription factor Slug and induce and sustain p38 mitogen-activated protein kinase (MAPK) phosphorylation [4].

During pEMT, polarized and differentiated epithelial cells acquire a reversible mesenchymal phenotype [6]; they underlie changes in morphology and give up their epithelial features and gene expression profile [7,8]. Cells undergoing pEMT release their lateral cell–cell contacts, their connection to the basal substrate, become motile, achieve collective invasion [4,6], and enforce stronger interaction with the extracellular matrix. These are typical features of the nonpolar mesenchymal phenotype [9]. During the process of pEMT, tumor cells switch their phenotype between epithelial and mesenchymal stages and achieve cellular plasticity [6]. The generation of mesenchymal phenotype, EMT, is considered a driver of metastasis. The reversal of EMT, transition to epithelial from mesenchymal (MET) phenotype, contributes to the metastatic outgrowth of disseminated cancer cells. Although pEMT is infrequent in tumor tissue, it significantly defines tumor progression in HNSCC patients [6]. These phenotypic changes are also manifested in treatment systems of HNSCC [10,11] since pEMT is responsible for tumor progression and RCT resistance even in the same tumor. Moreover, the surrogate pEMT factor Slug positivity [11] is a therapy-deciding predictive biomarker for a surgical preference [10]. The clinical significance of pEMT has been evidenced previously [12]; moreover, our previous finding that among the EMT transcription factors, only Slug has clinical significance was corroborated by single-cell sequencing [12]. Recently reviewed clinical data indicate worse survival for HNSCC patients with pEMT tumor cells [13].

Forty percent of HNSCC patients in Tyrol (Austria) obtain concomitant therapy, commonly consisting of cisplatin in combination with radiotherapy. The process of pEMT is the major reason for cisplatin resistance in HNSCC. The signaling pathways inducing pEMT are various, including NF-κB, Wnt, Notch, Hippo, JAK-STAT, AP-1, PI3K/AKT, and others [13,14,15]. Transforming growth factor beta-1 (TGF-beta-1) acts as an important external factor for the induction of pEMT and the Slug-positive phenotype in HNSCC [7,8,13,16,17]. Instead of the canonical TGF-beta pathway, which involves phosphorylated SMAD 2/3 proteins, pEMT is induced by the non-canonical TGF-beta pathway via p38 MAPK [18]. Our previous work provided evidence that the non-canonical TGF-beta/p38 pathway plays an essential role in the activation of the pEMT-related therapy resistance phenotype and downregulation of Krüppel-like-factor 4 (KLF4) in HPV-negative HNSCC [4,8]. KLF4 has been shown to be a potential inducer of mesenchymal-to-epithelial transition (MET) in HNSCC, and its expression is decreased in EMT cells in the presence of TGF-beta-1.

In HNSCC, especially the predominant p38 MAPK isoforms p38α and p38δ significantly promote the malignant phenotype of the tumor cells [19]. Furthermore, pospho-p38 was significantly increased detected in poorly differentiated HNSCC tissue samples compared to the normal oral squamous epithelium and HNSCC in the initial stage [20]. 

The p38 MAPK signaling pathway is a multi-tiered cascade characterized by sequential protein phosphorylation events. p38 protein (MAPK; α, β, δ, and γ isoforms) belongs to the subfamily of stress-activated MAP kinase family that is phosphorylated by various environmental stresses, pro-inflammatory cytokines, growth factors, and DNA damage agents [21,22]. p38 MAPK can be located in both the cytoplasm and the nucleus, depending on its de- and phosphorylated form.

Once activated, p38 either phosphorylates cytoplasmic targets or translocates into the cell nucleus, leading to the regulation of various transcription factors involved in cell cycle activity [23,24]. The phosphorylation-dependent nuclear export of p38 MAPK is a common phenomenon when tumor cells are exposed to chemotherapeutic agents. At the molecular level, members of the MEK family, such as MEK4 and MEK3/MEK6, stimulate p38 MAPK phosphorylation, which are themselves activated by several MAP-kinase kinase kinases (MAPKKKs) [21,24,25]. The nuclear translocation of p38 depends on its dephosphorylation and its export to the cytosol to receive further stimulations [23]. p38 MAPK phosphorylates a number of prominent downstream substrates, including MAPKAP-K2, the transcription-factor CREB, and cAMP-mediated activating transcription factor (TF) 2 (ATF2) [24,26,27].

To model the complexity of HNSCC, two different cell lines were used: UPCI-SCC090 cells and SCC-25 cells. UPCI-SCC090 cells represent characteristics of cisplatin-sensitive HPV-positive HNSCC [28], whereas SCC-25 cells display RCT-resistant properties of HPV-negative HNSCC [5]. 

The aim of our study was to determine the anti-tumoral effect of p38 MAPK inhibition in these cell lines using two specific p38 MAPK inhibitors, SB202190 and ralimetinib. SB202190 is a potent inhibitor of p38 mitogen-activated protein and selectively inhibits the prominent isoforms p38α and β [29,30,31]. Furthermore, SB202190-dependent p38 inhibition suppresses the activity of hypoxia-inducible factor α (HIF-1α), a transcription factor whose expression is triggered by hypoxia, a prevalent stress stimulus in many types of cancers, including HNSCC [30,32]. Ralimetinib (LY2228820 dimesylate) is an ATP-competitive p38 MAPK inhibitor that is currently in clinical trials [33,34]. Preclinical studies confirmed that ralimetinib inhibits p38 kinase activity as well as the secretion of IL-6 [33], a pro-inflammatory cytokine and external key regulator of pEMT in HNSCC [8,35]. In advanced cancer, the use of p38 MAPK inhibitors was found to have preliminary positive effects by inhibiting cancer cell growth and the reduction in antioxidant enzymes in cell lines and solid tumor xenograft models. Moreover, p38 MAPK inhibitors were recently described as important sensitizers for chemotherapeutic agents via JNK-mediated apoptosis [36,37].

In this study, we also confirm the results of previous papers that p38 inhibitor pre-treatment sensitizes cisplatin-resistant pEMT tumor cells to cisplatin.

## 2. Materials and Methods

### 2.1. Cell Lines

SCC-25 and UPCI-SCC090 cells were purchased from the German Collection of Microorganisms and Cell Cultures (DSMZ, Braunschweig, Germany). SCC-25 cells were cultured in DMEM/Ham’s F12 medium, and UPCI-SCC090 cells in EMEM medium. Both media were supplemented with 10% FBS, 2 mM L-glutamine, 100 units/mL penicillin and 100 μg/mL streptomycin (all from Pan-Biotech, Aidenbach, Germany), 1 mM sodium pyruvate and 1× MEM non-essential amino acids (Cat. Nr. P08-32100, Pan-Biotech).

### 2.2. p38-Signaling Inhibitors

SB202190 was purchased from R&D Systems (Minneapolis, MN, USA) and dissolved at 25 mM in DMSO. Controls to treatments with SB202190 received an equal volume of DMSO as solvent control. The final concentration of SB202190 was set to 25 µM [38]. Ralimetinib (LY2228820) was purchased from Selleckchem (Houston, TX, USA), and 1 mM stock solution was prepared in water. The final concentration of ralimetinib was set to 1.6 µM, which corresponds to the highest clinically measured blood plasma pharmacokinetic concentration [34]. Treatment with the p38 inhibitors was performed in two 72 h intervals, which is in line with previous literature [30].

### 2.3. Use of p38 Inhibitors and Cell Treatment

We used our previously published treatment schedule [4] with modifications. Seven thousand SCC-25 or 4 × 10^4^ UPCI-SCC090 cells per ml were plated on 6- or 96-well plates (day 1). After 24 h, all cells were supplied with serum-free DMEM-F12 (for SCC-25) or EMEM (for UPCI-SCC090) medium (serum proteins were replaced with bovine serum albumin (Serva, Heidelberg, Germany). From day 2 until day 5, cells were treated with serum-free medium optionally supplemented with 1 ng/mL TGF-beta-1 (R&D Systems, Minneapolis, MN, USA). From day 5 until day 8, cells were treated with serum-free medium optionally supplemented with 1 ng/mL TGF-beta-1 and/or with p38 inhibitors at the above-mentioned final concentrations. From day 8 until 11, cells were treated with serum-free medium optionally supplemented with p38 inhibitors. On day 11, the experiment was completed. The 96-well plates were used for MTT assay; the 6-well plates were used for immunocytometry, the Annexin V—FITC apoptosis assay, and cell cycle analysis or for Western blot. All experiments were repeated three times.

### 2.4. p38 Inhibitor Pre-Treatment before Cisplatin

Cisplatin was purchased from Selleckchem (Houston, TX, USA). A stock solution was prepared in deionized water (7 mM), which was prewarmed to 50 °C. Clinical levels of cisplatin treatment in vitro mean 10 µM of cisplatin final concentration, which corresponds to 100 mg/m^2^ cisplatin treatment in patients [39]. We performed all treatments with 10 µM cisplatin. The treatment duration was 24 h, and the effects of cisplatin were evident 3–4 days after the treatment [4].

Ten thousand SCC-25 cells per ml were plated on 96-well plates or in T-75 culture dishes (Sarstedt, Nümbrecht, Germany, 15 mL cell suspension per dish, day 1). After 24 h, all cells were supplied with serum-free DMEM-F12. From day 2 until day 5, cells were treated with a serum-free medium optionally supplemented with 1 ng/mL TGF-beta-1. From day 5 until day 8, cells were treated with serum-free medium optionally supplemented with 1 ng/mL TGF-beta-1 and/or with p38 inhibitors at the above-mentioned final concentrations. From day 8 until 11, cells were treated with serum-free medium optionally supplemented with p38 inhibitors. On day 11, all plates and dishes were supplemented with 10% serum-containing medium and 10 µM cisplatin. After 24 h, the cisplatin-containing medium was removed, and the plates and dishes were cultured with 10% serum-containing medium for an additional 72 h. The 96-well plates were used for MTT assay, the T-75 dishes for immunocytometry, Annexin V—FITC apoptosis assay, and cell cycle analysis. All experiments were repeated three times.

### 2.5. Inhibition of Acquired p38 Activity in Cisplatin-Surviving Cells

As we published earlier, SCC-25 cells, after TGF-beta-1-induced pEMT and cisplatin treatment, enabled sustained active phosphorylated p38 [4]. This initiated the experimental schedule that, at first, we brought SCC-25 cells to pEMT with two cycles of TGF-beta-1 treatment; after that, we treated them with cisplatin, replated the cells and performed treatment with p38 inhibitors.

Ten thousand SCC-25 cells per ml were plated in T-75 culture dishes (day 1). After 24 h, all cells were supplied with serum-free DMEM-F12. From day 2 until day 5 and from day 5 until day 8, cells were treated with serum-free medium optionally supplemented with 1 ng/mL TGF-beta-1. On day 8, all dishes were supplemented with 10% serum-containing medium and 10 µM cisplatin. After 24 h, the cells were trypsinized, and both control and TGF-beta-1-pretreated cells were plated at 2 × 10^4^ cells/mL on two six-well plates and one 96-well plate. After 24 h, cells in the six-well plates and some wells of the 96-well plates were treated with p38 inhibitors in serum-free conditions for two 72 h periods. The 96-well plates were used for MTT assay, the T-75 dishes for immunocytometry, Annexin V—FITC apoptosis assay, and cell cycle analysis. All experiments were repeated three times. 

### 2.6. Viability Assay

In the 96-well plates, the first row (row A) was used cell-free as blank control. After completion of the treatment schedules, 10 µL of MTT labeling reagent (Cell Proliferation Kit I (MTT), Cat. Nr. 11 465 007 001, Roche Diagnostics, Mannheim, Germany) was added to all wells of the 96-well plates. The plates were incubated for 4 h at 37 °C in a humidified CO_2_ incubator. After this incubation, 100 µL Solubilization solution (Cell Proliferation Kit I (MTT), Roche, Mannheim, Germany) was added to each well. The plates were incubated overnight at 37 °C in a humidified CO_2_ incubator. Spectrophotometrical absorbance of the wells was measured using an Anthos 2010 microplate (ELISA) reader (Salzburg, Austria) and the Galapagos software (Version 1.1.2.0, Biochrom, Cambridge, UK). The formazan product was quantified at 550 nm; the reference wavelength was 660 nm. The 550 nm absorbance values were corrected with the reference wavelength values, and the blank values were subtracted from those of the cell-plated wells. The mean absorbance of control or, in the case of cisplatin treatment, of only cisplatin-treated wells was considered 100%. All treatments were related to 100% control or only cisplatin-treated values. Treatment-caused changes were presented in scatter or boxplot diagrams depending on if the data were normally distributed or not and were compared using the D’Agostino and Pearson normality test and one-way ANOVA in Graphpad Prism (Version 9.4.1, Graphpad Software, San Diego, CA, USA).

### 2.7. Flow Cytometric Assessment of Apoptosis and Cell Cycle

After completion of the treatment schedules, cells were collected from culture dishes or plates by trypsinization (Pan-Biotech). Cell death was identified using the Annexin V-FITC/Fixable Viability Dye (FVD) 660 assay (Annexin: Biolegend, San Diego, CA, USA, FVD, Cat. Nr. 65-0864-14, eBioscience, Invitrogen, Darmstadt, Germany). Since no reaction was detected with FVD-660, the red 660 nm channel was used for the detection of cell surface EMT maker podoplanin. After collection, cells were incubated for 15 min in 50 µL/10^6^ cells of fetal bovine serum (FBS, Pan-Biotech) for blocking purposes. After this, 500 µL/10^6^ cells PBS was added, and cells were centrifuged at 290 g for 5 min at 4 °C. The cell pellet was resuspended in 500 µL Annexin Assay Buffer (0.1 M HEPES, 0.14 M NaCl, 2.5 mM CaCl_2_). Five µL Annexin V-FITC (Cat. Nr. 640906) and 2 µL Podoplanin APC antibody (Cat. Nr. 337021) (both from Biolegend) were added for a 25 min incubation at room temperature. Cells were centrifuged as described above. The cell pellet was resuspended in 800 µL Annexin Assay Buffer, filtered through 20 µm cell strainers (Partec, Görlitz, Germany), and used for flow cytometry on a Beckman Coulter CytoFLEX^TM^ flow cytometer. Analysis of flow cytometry data was carried out using CytExpert 1.2 (Beckman Coulter, Brea, CA, USA). Forward and side scatter were acquired at blue laser 488 nm, and fluorescent channels at 488 nm blue laser and 638 nm red laser were used. For detection purposes, FITC channel and APC channel were used.

In addition to flow cytometry, cell morphological changes in SCC-25 cells were assessed to examine the effects of cisplatin, TGF-beta-1, and p38 inhibitor SB202190 using a Juli BR live video-microscope (Nanoentek, Seoul, Republic of Korea).

For cell cycle analysis, cells were collected from culture dishes or plates by trypsinization, washed in PBS, and centrifuged as described above. Cells were resuspended at 2 × 10^6^ cells in 1 mL ice-cold PBS. By gentle vortexing, 9 mL of 70% ethanol was slowly dropwise added to the cell suspension in a 15 mL polypropylene centrifuge tube, as suggested by BD Biosciences, Vienna, Austria. The fixed cells could be stored at 4 °C. Cells were centrifuged as described above, and the pellet was resuspended in 50 µL FBS and incubated for 15 min at room temperature. Five hundred µL of PBS/Triton X-100 staining solution per 2 million cells was added (PBS, with 1 µL Triton X-100 per ml solution and 1 mg/mL RNase (Sigma, Darmstadt, Germany)), containing 10 µL 7-AAD (BD Pharmingen, San Diego, CA, USA) and 10 µL cytokeratin-FITC (Beckman Coulter) per 2 million cells. The cell suspension was incubated at 37 °C for 30 min. Cells were centrifuged at 290 g for 5 min at 4 °C, and the pellet was resuspended in 800 µL PBS/Triton X-100 staining solution, filtered through 20 µm cell strainers, and used for flow cytometry on a Beckman Coulter CytoFLEX TM flow cytometer. Besides the FITC channel for cytokeratin, the PC5.5 channel was used for 7-AAD. Negative controls for all flow cytometry assays were generated using cells incubated with isotype IgGs conjugated with FITC, APC, or PC5.5 (Biolegend). Thresholds were defined by a maximal 5% reaction with the negative controls.

### 2.8. Western Blot

After the treatments were completed, the cells were scraped into RIPA buffer [4,7] and used for protein isolation and Western blot. Rabbit mono- or polyclonal primary antibodies against p-p38 MAPK (1:1000), p-ATF2 (1:1000), p-MAPKAP2-kinase (1:1000), and Slug (1:1000) were purchased from Cell Signaling Technology (Danvers, MA, USA). For signal detection, highly specific isotype matching secondary antibodies (anti-rabbit), conjugated with either peroxidase (Invitrogen, Darmstadt, Germany) or near-infrared (NIR) fluorescence marker (Li-cor Biosciences, Lincoln, NE, USA), (1:1000, 1:10,000 dilutions), were used in conditions recommended by the manufacturers. GAPDH protein levels were considered a loading control, using an anti-GAPDH antibody (dilution 1:5000) purchased from Abcam (Cambridge, UK). For GAPDH detection, fluorescence conjugates labeled anti-mouse IgG secondary antibody were selected, available from Li-cor. Azure C500 Western blot imaging biosystem (Azure Biotech, Houston, TX, USA) was used to visualize specific NIR fluorescence and chemiluminescence signals. Western blot bands’ optical density measurements were carried out as published before [4].

### 2.9. Data Analysis 

All experiments were repeated three times. Data sets were tested for normal distribution using the D’Agostino and Pearson omnibus normality test. Comparison of two data sets was carried out using Student´s t-test in case of normal distributed data or with Wilcoxon matched-pairs signed rank test in case of non-parametric data. Western blot samples were compared in pairwise analysis as inhibitor-treated versus untreated samples. Comparisons of more data sets were conducted by parametric ANOVA or non-parametric Kruskal–Wallis test. Statistical analysis was performed by SPSS Version 29 (IBM, Chicago, IL, USA) and by Graphpad Prism Version 9.4.1. (La Jolla, CA, USA).

## 3. Results

### 3.1. p38 MAPK Inhibitor SB202190 Downregulates Protein Levels of Slug in SCC-25 Cells

Recently, the p38 MAPK pathway was shown to regulate EMT-related protein expression in response to TGF-beta-1 stimulation [4,40]. According to our previous results, TGF-beta-1 and cisplatin treatment induced p38 phosphorylation and the activation of mesenchymal vimentin and Slug in SCC-25 cells, whereas this effect could not be observed in the HPV-positive cell line UPCI-SCC090 [4]. To determine the potential of p38 inhibitor SB202190 in HNSCC treatment, SCC-25, and UPCI-SCC090 cells were stimulated for different periods with TGF-beta-1 and SB202190-inhibitor (concentration 9–25 μM) and prepared for Western blot analysis.

Selective inhibition of p38 MAPK activity clearly increased p-p38 levels in both SCC-25 and UPCI-SCC090 cells (Figure 1A,B). In SCC-25 cells, TGF-beta-1 treatment increased p-p38 and Slug protein levels (Figure 1C). In UPCI-SCC090 cells, p-p38 was upregulated (Figure 1D), whereas Slug protein was not influenced by TGF-beta-1 exposure. Further, our results revealed that Slug levels were consistently decreased after p38 inhibition in both SCC-25 and UPCI-SCC090 cells.

### 3.2. Inhibition of p38 MAPK Signaling Pathway Induces Growth Reduction in the RCT Resistant Cell Line SCC-25

In order to find out the potential inhibitory effects of SB202190 on SCC-25 and UPCI-SCC090 cells, 7 × 10^3^/mL SCC-25 and 2.5 × 10^4^/mL UPCI-SCC090 cells were plated. The cells were treated with two cycles of 1 ng/mL TGF-beta-1; the second cycle of TGF-beta-1 treatment was combined with 25 µM SB202190. After the second TGF-beta-1 incubation and optional 25 µM SB202190 treatment, an MTT assay was carried out to investigate the viability of the cells. Pre-treatment with TGF-beta-1 alone and SB202190 in combination with TGF-beta-1 led to significant growth reduction compared to the control. Moreover, relative viability was significantly reduced in the combined TGF-beta-1 and SB202190-treated cells compared to TGF-beta-1 treated cells (Figure 2A). The presence of SB202190 inhibitor alone did not achieve a significant inhibition in their growth, suggesting that cells undergoing pEMT were negatively influenced by inhibition of the p38 signaling pathway.

Similar to SCC-25 cells, UPCI-SCC090 cells also demonstrated a significant growth reduction by TGF-beta-1 pre-treatment, but the percentage of the relative viability of cells treated with TGF-beta-1 and the SB202190 inhibitor was not significantly decreased compared to TGF-beta-1 pre-treatment alone (Figure 2B).

### 3.3. p38 MAPK Inhibition Increases Apoptosis and Sensitizes HNSCC Cells to Cisplatin Treatment

Taking a kinase inhibitor into consideration, we expected a reversible growth reduction role and not a direct apoptosis induction of the p38 inhibitor SB202190. To obtain sustained pEMT [4], we performed two cycles of TGF-beta-1 pre-treatment in SCC-25 culture and one cycle of SB202190 treatment overlapping with the second cycle of TGF-beta pre-treatment. We expected that (1) TGF-beta-1 enriched pEMT cells would be cisplatin-resistant, and (2) SB202190 treatment would render even the resistant pEMT cells to cisplatin sensitive. TGF-beta-1 pre-treatment, as shown before, reduced the cell growth of SCC-25 cells (Figure 2A). TGF-beta-1 pre-treated cells still survived cisplatin at 75.33 ± 2.18% viability. SB202190 reduced the ratio of surviving cells to 26.3 ± 0.92% and, in the case of TGF-beta pre-treatment, to 11.19 ± 0.31%. This is significantly lower (*p* < 10^−4^) than the effect of the SB202190 inhibitor on not TGF-beta-1-pretreated cells (Figure 3A). The efficacy of SB202190 was confirmed by FACS analysis after the following treatment regimen: cisplatin treatment, TGF-beta-1 pre-treatment before cisplatin, SB202190 followed by cisplatin treatment, TGF-beta-1 plus SB202190 pre-treatment followed by cisplatin treatment.

In SCC-25 cells, after the above-described treatments, apoptosis was evaluated with Annexin V-FITC combined by quantification of cell surface podoplanin expression. Podoplanin is considered a major pEMT-specific cell surface marker in HNSCC [13]. No staining was detected with 7-AAD or with fixable viability dye in unfixed cells, and they were also trypan-blue-negative after all treatment conditions. This made it possible to replace the y-axis in the scattergram (Figure 3B) from the necrosis marker to podoplanin as a pEMT marker [41]. The used podoplanin antibody reacted with the extracellular domain and was suitable for unfixed cells, offering a combination with the Annexin detection. This approach allowed the evaluation of apoptosis in cells in pEMT, which is a main focus of this work.

In the case of cisplatin single treatment, 30.42% of the tumor cells demonstrated a positive signal for Annexin (Figure 3B). The apoptotic signal was significantly higher than in cells pre-treated with TGF-beta-1 (18%, Figure 3C). In the SB202190 and cisplatin treatment combination, 39.15% of tumor cells tended toward apoptosis (Figure 3D). Similar was the apoptotic effect in TGF-beta-1 pre-treated cells followed by SB202190 and subsequently cisplatin (34.52%, Figure 3E). As expected, TGF-beta-1-induced stable pEMT resulted in cisplatin resistance (18% apoptosis vs. 30.42%; Figure 3C vs. Figure 3B). This tendency was reverted by the p38 inhibitor, allowing as much apoptosis in the stable pEMT culture as in the cisplatin + SB202190—treated one (34.52% vs. 39.15% apoptosis).

### 3.4. p38 MAPK Inhibitor Targets Tumor Cells Surviving Cisplatin

As detailed before, we confirmed that inhibition of p38 MAPK led to sensitization to cisplatin-induced cell death in the cisplatin-resistant HNSCC cell line SCC-25. To determine the role of p38 MAPK inhibition in cisplatin-surviving tumor cells, differences between cisplatin alone (control) and cisplatin plus SB202190 treated cells as well as TGF-beta-1-stimulated before cisplatin and SB202190-treated after cisplatin was evaluated by MTT assay and FACS analysis. In both TGF-beta-1-pretreated and only cisplatin-pretreated conditions, the subsequent SB202190 treatment cycles significantly (*p* = 10^−3^–10^−4^) reduced the fraction of surviving cells (Figure 4A). Annexin V-FITC was used to identify and quantify apoptotic cells, and pEMT tumor cells were detected using podoplanin antibody. The apoptotic cells in the cisplatin-treated sample amounted to 38.43% (Figure 4B), and two cycles of SB202190 after cisplatin treatment increased the apoptotic cell ratio up to 50.94% (Figure 4D). Further, our results indicate that the podoplanin/Annexin double-positive cells were higher in the samples treated with cisplatin plus p38 MAPK inhibitor compared to cisplatin alone (8.08% vs. 4.41%) (Figure 4D vs. Figure 4B). TGF-beta-1 pre-treatment before cisplatin demonstrated the highest population of surviving podoplanin positive cells (4.14%) and, in accordance with this, the lowest apoptosis rate (25.12%) (Figure 4C). Importantly, the results provide evidence supporting that SB202190 potentiates cisplatin-induced apoptosis (58.29%) in TGF-beta-1 pre-treated cells (Figure 4E). Moreover, SB202190 reduced the percentage of TGF-beta-1-induced Annexin-negative pEMT cells from 4.14% to 0.80% (Figure 4C,E). Finally, our data revealed that p38 MAPK inhibition cooperates with cisplatin treatment and promotes apoptosis in podoplanin-labeled pEMT tumor cells (9.74% vs. 2.56%) (Figure 4E vs. Figure 4C).

Phase contrast images were taken before the cells were prepared for flow cytometry. After cisplatin treatment, plates contained giant cells with multiple nuclei; other cells showed epithelial morphology. Exposure to TGF-beta-1 changed cells into a mesenchymal-like elongated cell morphology. In the case of co-treatment with SB202190, cisplatin-resistant giant cells and apoptotic bodies were present. p38 MAPK inhibition was sufficient to suppress the proliferation of tumor cells surviving cisplatin and/or even induced cell death, characterized by apoptotic morphologies (Figure 5).

### 3.5. Clinically Relevant p38 MAPK Inhibitor Ralimetinib Causes Apoptosis without Cisplatin Co-Treatment in HNSCC

The previous results revealed that SB202190 was able to sensitize the cisplatin-resistant pEMT cells for cisplatin-induced cell death. Moreover, cells surviving cisplatin died upon SB202190 treatment. These findings suggested that the p38 MAPK inhibitor might have its own pro-apoptotic effect, independently from cisplatin. Apoptosis was investigated by Annexin/FITC levels and DNA degradation. The results of both methods were interpreted.

To determine whether two cycles of SB202190 treatment induced apoptosis in the RCT-resistant cell line SCC-25, the fluorescence detection levels of Annexin/FITC were examined by flow cytometry analysis based on the following treatment schedule: control; SB202190 treatment; TGF-beta-1 stimulation; and TGF-beta-1 and SB202190 treatment together. SB202190 did not increase Annexin positivity compared to the control group. We observed an increased Annexin signal by TGF-beta-1 alone (14.41%), which is in line with other HNSCC studies [18,42], where TGF-beta-1 was found to induce apoptosis (Figure 6A).

Our data revealed that the presence of TGF-beta-1 and SB202190 (even at a decreased 10 μM concentration) clearly increased the percentage of Annexin-positive cells (17.41%) compared to the control group (8.25%) (Figure 6A).

SB202190 is a selective inhibitor of p38 mitogen-activated protein kinase with a potent experimental background [30,38]. In fact, SB202190 has been used in various in vitro and in vivo systems, but it could not proceed in clinical trials. Alternatively, we tested ralimetinib, an effective p38 inhibitor, used for the treatment of metastatic breast cancer and other advanced cancer types in phase II clinical trials, alone or in combination with currently available therapeutic modalities [34,43]. The experimental design was set up as described before. SB202190 was replaced by ralimetinib (concentration 1.6 µM). As represented in Figure 6B, the Annexin positivity induced by ralimetinib was higher compared to the control. We reproduced the above-described results with TGF-beta-1 treatment alone. TGF-beta-1 and ralimetinib co-treatment achieved the highest Annexin positivity (Figure 6B).

Cell cycle distribution of control, SB202190 (C) or ralimetinib (D), TGF-beta-1 and TGF-beta-1/p38 inhibitors combined-treated cells presented on linear scaled 7-AAD signal area histograms of SCC-25 cells. The 7-AAD histograms-based sub-G1, G1, S, G2-M phases are indicated in the figures.

Since apoptosis might be induced without changes in Annexin level, and Annexin alone is not sufficient to claim apoptosis, we ethanol-fixed and 7-AAD stained the SCC-25 cells treated with the above schedule. The resulting DNA histograms allowed us to gain data on DNA degradation and cell cycle distribution. Cells with DNA content lower than the main diploid population were interpreted as apoptosis, which were in the sub-G1 phase representing DNA degradation. This phase was at 3.13–3.76% in the controls, 39.25% in the SB202190-treated and 14.51% in the ralimetinib-treated cells, 10.59–11.20% in the TGF-beta-1-treated cells, in the TGF-beta-1- SB202190-combined treated cells it was 18.10%, whereas, in the TGF-beta-1- ralimetinib-combined treated cells it was 7.88% (Figure 6C,D). The cells recruited in the cell cycle were detected in the G1, S, and G2/M phases based on the DNA content. The distribution among the phases of the cell cycle was also influenced by SB202190 in both single-treated and combined with TGF-beta-1-treated form, as it pushed the cell cycle to less DNA-containing phases: G1 arrest and G1-sub-G1 arrest. G1/S arrest and induction of sub-G1 phase (less G2 cells) was also characteristic of TGF-beta-1 treatment. Interestingly, TGF-beta-1 combined with ralimetinib was characterized by G2/S arrest; cells with more DNA content were increased, whereas ralimetinib alone induced G1/S, G1/sub-G1 swift. In this relation, the sub-G1 phase was not characteristic of apoptosis in the cells treated with TGF-beta-1 combined with ralimetinib, while in these cells, higher DNA-containing cell cycle phases (S and G2) were also arrested by ralimetinib (Figure 6D). This was combined with increased Annexin reaction (Figure 6B).

### 3.6. Ralimetinib Targets the Downstream Protein ATF2 in HNSCC

The biomechanistic pathways and downstream targets of p38 inhibitors in HNSCC are mainly unclear. This lack of knowledge raised the approach to treat SCC-25 cells with p38 inhibitors to explore the protein phosphorylation of the downstream targets p-ATF2 (Thr69/71) and MAPKAP-2 (Thr222). Interestingly, ralimetinib reduced the p-ATF2 (Thr69/71) phosphorylation, which appears to be maximally decreased in TGF-beta-1 stimulated and ralimetinib co-treated cells (Figure 7A,B). Phosphoprotein levels and changes in the downstream target MAPKAP-2 were not detected, independent of SB202190 or ralimetinib usage. SB202190-mediated inhibition of p38 MAPK did not significantly affect p-ATF2 phosphoprotein levels (Appendix A).

## 4. Discussion

The p38 MAPK pathway regulates several cellular functions, including cell differentiation, apoptosis, and cell-cycle activity [21,44], while in HNSCC and other tumor types, activation of p38 MAPK is clearly associated with oncogenesis and tumor progression [20,45], contributing to the production of pro-inflammatory cytokines, such as IL-6, and tumor necrosis factor α (TNF-α) [46].

Numerous studies demonstrated that the inhibition of p38 MAPK isoforms α and β can constrain tumor progression and invasion [19]. In this HNSCC in vitro study, the use of p38 MAPK inhibitors was found to have preliminary positive effects by reducing cancer cell growth and enhancing the sensitivity to the commonly applied chemotherapeutic agent, cisplatin.

Biological activation of p38 MAPK occurs via dual phosphorylation at Thr180 and Tyr182, mediated by the upstream kinases MKK3, SEK-1/MKK4, and MKK6 [25,47]. As represented in Figure 1, SB202190 treatment increased p-p38 MAPK levels in SCC-25 and UPCI-SCC090 cells. SB202190 blocks p38 MAPK catalytic activity, but the p38 MAPK remains phosphorylated. In this case, the kinase cascade will be interrupted, and p38 MAPK remains frozen in the phosphorylated form.

p38 MAPK is activated in the course of pEMT [4], most importantly, by a non-canonical TGF-beta-1 pathway, where the downstream signaling target from p38 MAPK is ATF2 [26]. ATF2 belongs to the first reported p38 MAPK downstream substrates, activated by phosphorylation of Thr69 and Thr71 [26,48]. Deregulations of ATF2 are implicated in tumorigenesis, reflecting its role in chromatin remodeling and cell cycle progression.

Inhibition of p38 MAPK phosphorylation might influence more downstream targets in addition to ATF2 and can be effective to induce significant tumor cell apoptosis at therapeutically relevant levels [30,49]. Nevertheless, a multi-targeted hit might be related to increased side effects, where anti-tumor and anti-inflammatory effects must be patient-specifically personalized and weighted [50,51].

Our results, in agreement with our previous study evidence that the non-canonical TGF-beta-1 signaling pathway over p-p38 might be responsible for the activation of the RCT resistance factor Slug [4]. Inhibition of this pathway using p38 inhibitors also reduces the Slug levels. This might occur over the downstream inhibition of p-ATF2 [52,53].

According to our experimental data, we observed various p-p38 MAPK phosphoprotein levels in HNSCC cell lines and in HNSCC patients using antibodies against p-p38 MAPK phosphoprotein in immunohistochemistry (not shown). Thus, HNSCC patients with p-p38 MAPK-positive tumor cells in the tumor tissue and p-p38 MAPK-positive chronic inflammation in the tumor site might be suitable for p38 MAPK inhibitor therapy.

Specific p38 MAPK inhibitors are considered beneficial for the treatment of several diseases, such as chronic obstructive pulmonary disease (COPD) [54], coronary artery disease, arteriosclerosis, facioscapulohumeral muscular dystrophy, autoimmune diseases [46], and other immunological dysregulations. The main track for p38 MAPK inhibition is anti-inflammatory activity.

Currently, seven ongoing clinical trials target p38 MAPK for cancer therapy [22,50]. Three p38 MAPK inhibitors, including ralimetinib [43], SCIO-469 [55], and LY3007113 [51], are tested for the treatment of adult glioblastoma [56], ovarian cancer, multiple myeloma, metastatic breast cancer and other advanced cancer types, alone or in combination with currently available therapeutic modalities. All information was obtained from clinicaltrials.gov [57]. Several studies revealed the promising approach in anti-tumor therapy by combining conventional chemotherapy with the p38α and p38β inhibitor ralimetinib, demonstrating antiangiogenic and pro-apoptotic effects in tumor patients and in cell lines [22,24].

Our experimental data support the role of p38 inhibitors as sensitizers for cisplatin therapy and suggest ralimetinib as a potential co-treatment to standard RCT for patients with significant Slug immunohistochemical positivity or pEMT tumor profile. At the same time, our results indicate that the p38 inhibitors on other HNSCC cells, such as the HPV-positive oropharynx squamous cell carcinoma (OSCC) cell line UPCI-SCC090, are less effective, not inducing significant reproducible apoptosis. If the p38 inhibitors could be validated for a targeting combination of pEMT HNSCC, which is RCT and PD(L)-1-based immunotherapy (immune checkpoint blockade (ICB)) resistant [58], it could be the first valid therapeutic approach for this tumor. Our suggestion is the development of ralimetinib for particularly RCT- and ICB-resistant pEMT HNSCC tumors.

As we published before, a repeated TGF-beta-1 treatment was required for stable and reproducible pEMT in SCC-25 cells [4,7], which knowledge was also used in this work. In fact, not only the TGF-beta-1 treatment was required in more cycles, but also the p38 inhibitors were given in more cycles to obtain reproducible induction of tumor cell apoptosis. There are clinical reports, as recently reviewed by Wakasugi et al. [59], on the effectiveness of reapplied therapies, which are, as shown here, relevant for in vitro treatments with p38 inhibitors. In addition, repeated TGF-beta-1 and p38 MAPK inhibitor treatments might also induce cellular senescence, which is relevant to discuss here.

Similarly to oxidative stress, chemotherapeutic agents and signaling pathways, such as TGF-beta-1 [60,61] and the p38 MAPK [62] cascade induce senescence of tumor cells. As critical modulators of senescence serve the p21 protein and the complex interplay of the tumor suppressor proteins p53 and Rb, two frequent inactivated proteins in HNSCC [16]. Key shared modulators of pEMT and senescence are Zeb1, Zeb2, and Slug over the p16-induced degradation of Rb and mutant p53 protein [63,64]. Senturk et al. [61] identified the TGF-beta signaling activity for a robust cellular senescence response over the upregulation of p21^Cip1^ and p15^Ink4b^ and achieved permanent G1 arrest. Based on our cell cycle analysis TGF-beta-1 treatment resulted in G1 arrest, but also in DNA degradation (Figure 6C,D). These data indicate that TGF-beta-1 might induce senescence and apoptosis at the same time; further evidence for senescence with p16 antibody via Western blot analysis could not be achieved. As recently reviewed by Martini and Passos, mitochondrial dysfunction (MD) is a hallmark of cellular senescence, which is involved in senescence-related growth arrest [65].

In pEMT cells of HNSCC, MD was published by us before [3]. In cells suffering MD downregulation of mitochondrial proteins or oxidative phosphorylation and upregulation of glycolysis and glycolytic pathways interact with pEMT transcription factors. In our previous study, HNSCC cell lines were treated with TGF-beta-1 and mitochondrial inhibitors. Increased expression of pEMT transcription factors, emergence of lipid droplets, and morphological pEMT characteristics were observed both after TGF-beta-1-treatment and after mitochondrial inhibitors [3].

Since the biological actions of ralimetinib and SB202190 in HNSCC were found to be different, the investigation of side effects at the cellular level is a possible aim for a follow-up study, especially considering ralimetinib, which might have a clinical potential for actually not curable pEMT HNSCC. In contrast, SB202190 has no further clinical application, according to previous works [22,66]. As presented before, we tried to exemplify more possible variants of HNSCC, considering the therapy-resistant pEMT tumor cells, which are represented by SCC-25 cells, as well as the therapy-responsive HPV-positive OSCC cell line UPCI-SCC090. Nevertheless, cell lines do not fully represent heterogeneity in tumor tissue, which is a limitation of this study. A model representing better the pEMT and non-pEMT tumor cells in the context of the tumor microenvironment (TME) is required for bridging ralimetinib for clinical use. In particular, the anti-inflammatory potential of ralimetinib, which, as shown here, is completed by the anti-tumor effect on pEMT tumor cells, would indicate its relevance in the context of therapy-resistant HNSCC, where the inflammatory TME confirms the pEMT by protein stabilization of Slug. In the advanced stage of HNSCC, the p38 inhibition could reduce both the inflammation in the tumor microenvironment and the therapy resistance of tumor cells. In the future, we aim to test the clinically relevant p38 MAPK inhibitor ralimetinib in patient-derived organotypic models.

## Figures and Tables

**Figure 1 biomedicines-11-03301-f001:**
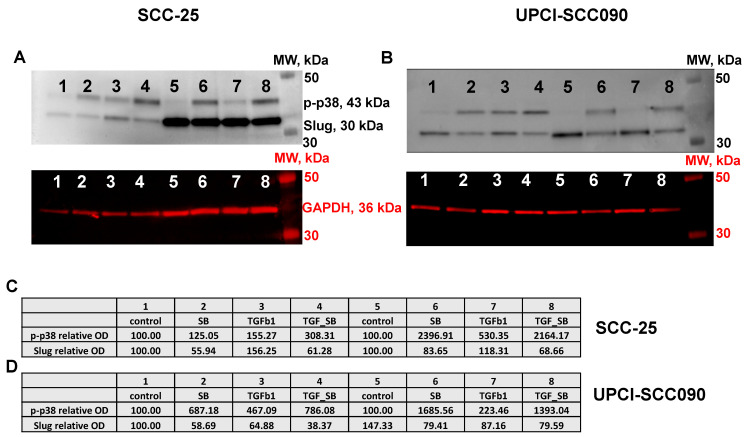
Downstream regulation of pro-EMT TF Slug by p38 MAPK inhibitor in HNSCC cell lines. SCC-25 (**A**) and UPCI-SCC090 cells (**B**) were treated with TGF-beta-1, and with SB202190. Phospho-p38 and Slug optical densities were normalized with those of GAPDH. The normalized optical densities (OD) of each experiment (1–4 or 5–8) were related to the controls (sample 1 or 5), where the OD of the controls was considered 100%. The relative optical densities of SCC-25 (**C**) and UPCI-SCC090 (**D**) cells for p-p38 and Slug are shown in tables.

**Figure 2 biomedicines-11-03301-f002:**
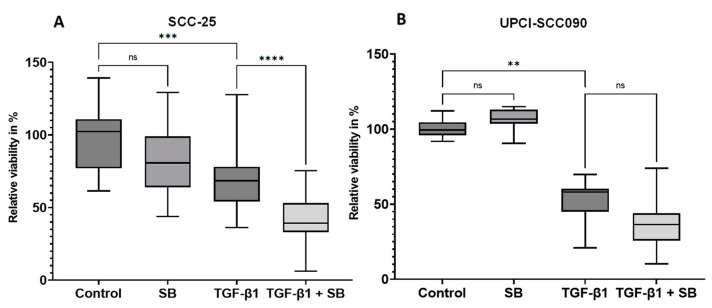
Involvement of p38 MAPK inhibition in the reduction in tumor cell growth in SCC-25 and UPCI-SCC090. MTT viability results in SCC-25 (**A**) and UPCI-SCC090 (**B**) cells (n = 24 per dataset) in relation to the mean of control cells, which was considered 100%. ****: *p* < 10^−4^, ***: *p* < 10^−3^, **: *p* < 0.01; ns: not significant, *p* > 0.05.

**Figure 3 biomedicines-11-03301-f003:**
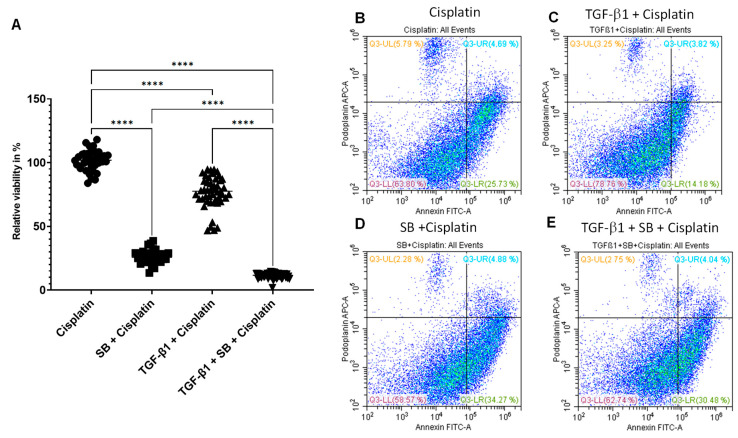
p38 MAPK inhibition enabled cisplatin-induced apoptosis in pEMT HNSCC tumor cells. (**A**) MTT viability results (n = 48 per dataset) in relation to the mean of only cisplatin-treated cells, which was considered 100%. Medians with 95% confidence intervals are presented on the graph. ****: *p* < 10^−4^. (**B**–**E**) Flow cytometry of Annexin V-FITC (X-axes) and podoplanin-APC (Y-axes) presented in scattergrams of only cisplatin-treated (**B**); TGF-beta-1-pretreated before cisplatin (**C**); SB202190 pre-treated before cisplatin (**D**) and TGF-beta-1- SB202190-combined-pretreated before cisplatin-treated samples (**E**) of SCC-25 cells. Lower left quadrants: double-negative cells; upper left quadrants: podoplanin-positive and Annexin-negative pEMT cells; lower right quadrants: Annexin-positive and podoplanin-negative apoptotic epithelial cells; upper right quadrants: double-positive apoptotic pEMT cells. Representative flow cytometry scattergrams of three repeated experiments.

**Figure 4 biomedicines-11-03301-f004:**
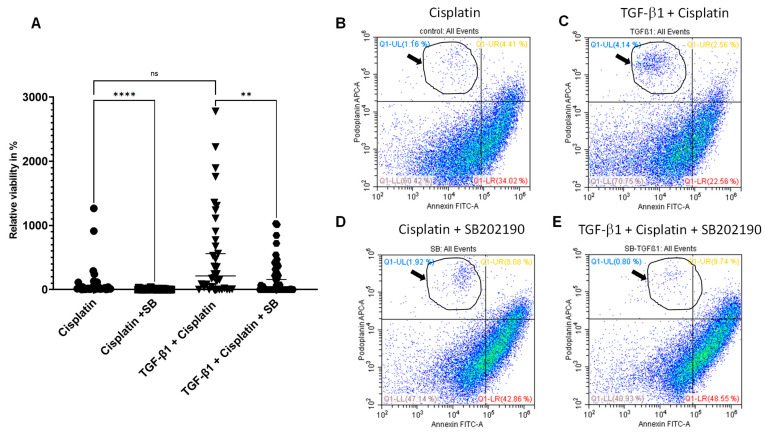
Inhibition of p38 MAPK pathway after cisplatin resulted in enhanced apoptotic effects in pEMT tumor cells. SCC-25 cells were treated with TGF-beta-1 followed by cisplatin treatment. Cisplatin-treated cells received two cycles of SB202190 after that. (**A**) MTT viability results (n = 48 per dataset) in relation to the mean of only cisplatin-treated cells, which was considered 100%. ****: *p* < 10^−4^, **: *p* < 0.01, ns: not significant, *p* > 0.05. (**B**–**E**) Flow cytometry of Annexin V-FITC (X-axes) and podoplanin-APC (Y-axes) presented in scattergrams in only cisplatin-treated (**B**); TGF-beta-1-pretreated before cisplatin (**C**); cisplatin plus two cycles SB202190-treated (**D**) and TGF-beta-1-pretreated before cisplatin and SB202190 treated after cisplatin samples of SCC-25 cells (**E**). Lower left quadrants: double-negative cells; upper left quadrants: podoplanin-positive and Annexin-negative EMT cells; lower right quadrants: Annexin-positive and podoplanin-negative apoptotic epithelial cells; upper right quadrants: double-positive apoptotic pEMT cells. TGF-beta-1-induced annexin-negative pEMT cells disappear if the sample was treated with SB202190 (lassos labeled with arrows). Representative flow cytometry scattergrams of three repeated experiments.

**Figure 5 biomedicines-11-03301-f005:**
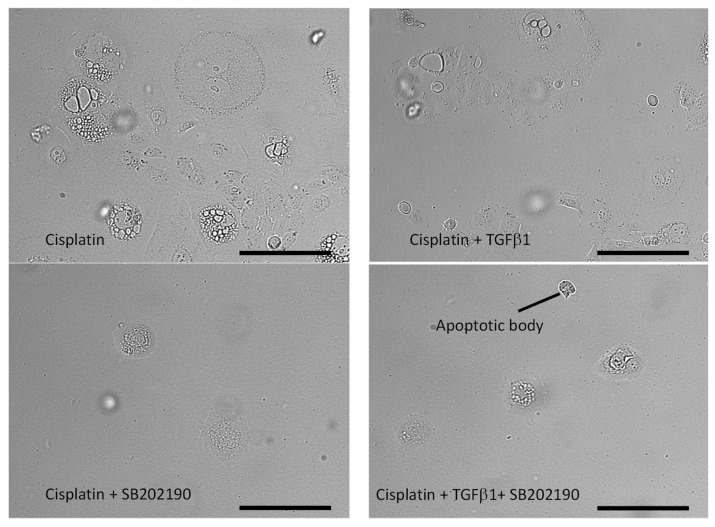
Cell morphology changes in SCC-25 cells after cisplatin and SB202190 treatment. Images taken directly before FACS analysis after cisplatin treatment and 2 cycles of SB202190. After cisplatin, not only the giant cells with multiple nuclei but also epithelial cells were present. TGF-beta-1 pre-treatment resulted in a similar image of epithelial cells and scattered pEMT cells. In the case of additional treatment with SB202190 directly after cisplatin and also in the case of TGF-beta-1 pretreated cultures, plates demonstrated single giant cells containing lipid droplets in combination with apoptotic bodies without evidence for intact epithelial cells. Bars: 100 µm.

**Figure 6 biomedicines-11-03301-f006:**
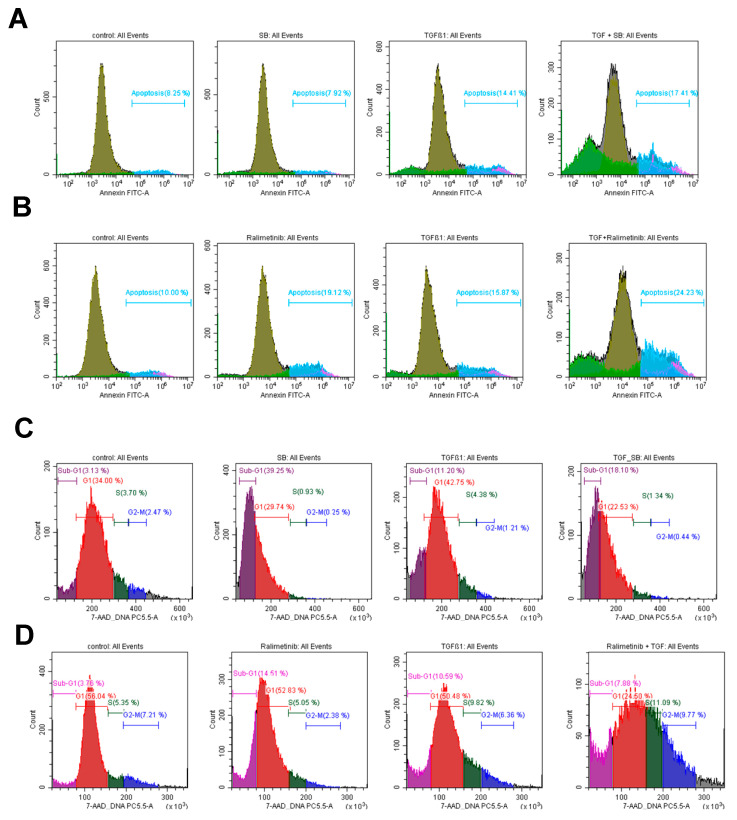
Comparison of the apoptosis induction effects of SB202190 (**A**,**C**) and ralimetinib (**B**,**D**) in SCC-25 cells. After the cell treatments, Annexin-V-FITC-based apoptosis (**A**,**B**) and 7-AAD-staining-based cell cycle analysis (**C**,**D**) were carried out via flow cytometry. The figures represent typical flow cytometry histograms, examples of three repeated experiments.

**Figure 7 biomedicines-11-03301-f007:**
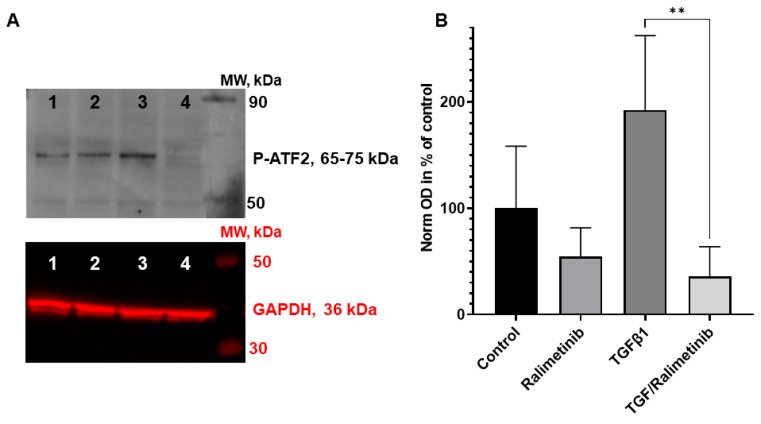
SCC-25 cells were treated with control conditions (1), ralimetinib (2), TGF-beta-1 (3), and TGF-beta-1 plus ralimetinib (4) (**A**). p-ATF2 protein was analyzed by Western blot in four independent experiments. Immunoblotting reactions were visualized by chemiluminescence. GAPDH was used as a loading control and was detected with NIR fluorescence. Phospho-ATF2 optical densities were normalized with those of GAPDH. The normalized optical densities (OD) in all experiments were related to the mean of the four controls, where this mean OD of the controls was considered 100%. The relative optical densities of control and treatments (n = 4 in all cases) are presented on a bar chart (**B**). **: *p* < 0.01.

## Data Availability

All original research data are included in the submitted article and in the Appendix A.

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
