# Peer review of "p38 Mitogen-Activated Protein Kinase Inhibition of Mesenchymal Transdifferentiated Tumor Cells in Head and Neck Squamous Cell Carcinoma"

_biomedicines, 2023, doi:10.3390/biomedicines11123301_

Round 1
Reviewer 1 Report
Comments and Suggestions for Authors
In the introduction authors need to better define the background data from which they started. The results of their previous work are not clearly resumed. They also are encouraged to better describe the role of partial EMT in cancer citing other reviews (i.e. ONCOGENE 2021, 40(32) 5049-5065; Developmental Cell 2021, 56(3), 3174-3176).
The material and methods section is sometimes too extended. Some experimental indications can be omitted (i.e. centrifuge details).
Regarding the use of Annexin V flow cytometry analysis I'm not convinced by the substitution of APC channel (live dead marker) with podoplanin antibody. How the authors can be sure that the APC signal is always negative for dead cells? They do not present a dot plot showing that. I think that it is more appropriate to use an antibody for podoplanin in a different fluorescence and before analysing annexin V+podoplanin+ cells to exclude dead cells in the gating strategy. Moreover, the authors presented graphs of MTT assay: they plot the relative viability, how they can say that cells analysed for FACS are totally viable?
I'm not really convinced about the dot plots in figure 3 and 4. The authors need to present a dot plot showing the staining without Annexin to be sure of the negative /positive separation.
In the process of EMT the production of cytokines such TNfalfa (doi: 10.1159/000489990) is important and correlates with the expression of SLUG and others EMT proteins (Snail, Twist). Do the authors have measured the concentration of such cytokine in the cell supernatant during their experiments? They also referred that IL-6 secretion (line 103) is inhibited by ramlimetinib but they do not report this analysis in their treatments.
It is known that treatment with TGFb in some tumor cell lines (doi: 10.1002/hep.23769.) induces cellular senescence. Do the authors can investigate this aspect in their experiments? Since they presented apoptosis and cell cycle it is important to point attention also at this cellular phenomenon. They can use immunostaining with galactosidase assay or WB for p21, p16 or related protein typical of senescent phenotype. Indeed, cells pretreated with TGFbeta presented a reduced percentage of apoptosis: it is possible that they are not apopototic but are senescent? In the figure 5 they showed giant cells with lipid droplets and some senescent cells look the same. This is something that authors need to discuss.
An in vivo experiment even if I know that is demanding, would increase the importance of the paper.
Author Response
Please find the point-by-point response to the reviewer’s comments in the attached pdf file.

Reviewer 2 Report
Comments and Suggestions for Authors
Some suggestions to better serve our readers:
1. The authors are encouraged to design experiments to validate the cell death mechanisms described in the article.
2. The author needs to conduct rescue experiments.
3. The validity of key conclusions should be substantiated through diverse experimental validations.
Author Response
Answers to the Comments of Reviewer 2.
Comment 1: The authors are encouraged to design experiments to validate the cell death mechanisms described in the article.
Answer: Cell death was recognized by reduction of cell numbers, whereas, in TGF-beta-1 treatment conditions also senescence could have been induced. Apoptosis was found by Annexin V detections as well as by DNA degradation. Nevertheless, detection of apoptosis was surprising for a p38MAPK inhibitor, although there are references for this in the literature (Nemoto et al. doi.org/10.1074/jbc.273.26.16415; Navas et al. doi.org/10.1038/sj.leu.2404200). Both of these references are cited in the Introduction and in the Discussion. Since clinically relevant p38MAPK inhibitors, especially ralimetinib or other newer compounds might achieve inhibition of signaling pathways with more downstream consequences leading even to cell death, more details of the achieved cell death induction will make sense to investigate.
Comment 2: The author needs to conduct rescue experiments.
Answer: As mentioned above, p38MAPK inhibitors might achieve inhibition of signaling pathways with more downstream consequences, rescue experiments might be also complex, and they are out of the scope of the current manuscript. Nevertheless, this approach will help to understand the significance of the MAPK inhibition for targeting the therapy resistant EMT cells in the future.
Comment 3: The validity of key conclusions should be substantiated through diverse experimental validations.
Answer: The manuscript is designed to achieve complementary methodology and confirm the outcomes also by results coming from other approaches, as DNA degradation and Annexin for apoptosis. MTT assay was confirmed by cell counting before moving on to flow cytometry, which was done in parallel with MTT-assay, furthermore, using Juli BR (Nanoentek, Seoul, South Korea) videomicroscope and phase contrast imaging system the confluence coverage of the cultures was also investigated. Necrosis was not detected by the used FVD using flow cytometry as well as trypan blue staining also revealed less than 5% positivity in all cases.
Reviewer 3 Report
Comments and Suggestions for Authors
The authors of the manuscript elaborated the potential of p38 signalling inhibition to limit head and neck cancer cell growth by targeting subpopulation of cells underwent partial EMT. Since the presence of these cells is tightly connected with establishment of radio- and chemo-resistance, downregulation of p38 lead to restoration of phenotype sensitive to standard therapeutic protocols. These data strongly support the implementation of p38 inhibition in high grade tumor treatment. Namely, during the tumor progression coordinated with continuous and dynamic stem phenotype establishment at multiple levels and at least partly realized through EMT, resulted in the presence of cell fraction unresponsive to conventional protocols. Thus, the work presented in this manuscript is multiply valuable, even if it is performed in vitro and in 2d cultures. I would like to underline excellent design of experiment, enabling the in vitro simulation of the complex process of tumor expansion and high grade tumor formation. Using this platform the authors gave the important and fundamentally relevant insight in the essential processes of diseases progression, and indicated strategies that can improve the protocols in advanced cancer treatment.
However, there are some imperfections in the manuscript that need to be corrected.
A general observation is that the legends for all figures are inadequately written, making it difficult to follow.
Each figure should have a title with the main conclusion of the presented results. Many methodological details are needlessly included in the legend, for example in Figure 1- Cells were scraped in RIPA buffer and 10 µg protein samples were separated and immunoblotted against phospho-p38 (Thr180/Tyr182) and Slug, GAPDH was used as loading control. Phospho-p38 (43 kDa) and Slug (30 kDa) were detect…etc) with chemiluminescence, GAPDH (36 kDa). This shouldn’t be a part of the legend.
The result explanations are not recommended to be a part of the legend (for example, Fig 6- In this evaluation method (Annexin) ralimetinib seemed to be more effective compared to low (10 µM) concentration of SB202190…. Etc) The comments of the results are exclusively part of the Result and Discussion section, and in Figure legend, only instructions for figure understanding are relevant.
Discussion is very clearly written. At the end the authors commented the that cell lines do not fully represent heterogeneity in tumor tissue, which is a limitation of this study and that they are planning in the future to test the clinically relevant p38 MAPK inhibitor ralimetinib in patient derived organoids. I suggest to authors to extend the discussion in terms of the possible consequences of p38 inhibition on tumor microenvironment in advanced stage, knowing the importance of intercellular communication as well as metabolic specificities of constituents and their interplay. pEMT cells possesses character of stemness and mostly underwent to metabolic shift to a glycolytic mode of energy production even in the presence of oxigen. It would be very beneficial to predict some of the consequences that abolished activity of these cells through p38 suppression might have on TME and subsequently, whole process of disease progression and therapy response.
Minor corrections:
126 and 127. Please rephrase.
141/142. Please rephrase. The term of biological repeats is unclear. It is enough to tell that all exp. were repeated tree times
144-145 incomplete
145- Rephrase the title
199- Title should be changed. Maybe like: Flowcytometric assessment of……
240-242. I suggest to write this as a the separate paragraph in Material and methods section, since this is microscopically evaluation, and it is not a part of flowcytometry, even the authors use the cells prepared for it to visually explore them.
After revision, I support the acceptance of this manuscript for the publication in Biomedicines.

Moderate language revision.
Author Response
Answers to the Comments of Reviewer 3.
Comment 1: A general observation is that the legends for all figures are inadequately written, making it difficult to follow.
Answer: Authors completely agree with this point. The figure captions were revised and clearly reduced as suggested by the Reviewer.
Comment 2. Each figure should have a title with the main conclusion of the presented results. Many methodological details are needlessly included in the legend, for example in Figure 1- Cells were scraped in RIPA buffer and 10 µg protein samples were separated and immunoblotted against phospho-p38 (Thr180/Tyr182) and Slug, GAPDH was used as loading control. Phospho-p38 (43 kDa) and Slug (30 kDa) were detect…etc) with chemiluminescence, GAPDH (36 kDa). This shouldn’t be a part of the legend.
Comment 3. The result explanations are not recommended to be a part of the legend (for example, Fig 6- In this evaluation method (Annexin) ralimetinib seemed to be more effective compared to low (10 µM) concentration of SB202190…. Etc) The comments of the results are exclusively part of the Result and Discussion section, and in Figure legend, only instructions for figure understanding are relevant.
Answer: All figure legends are corrected as suggested.
Comment 4. Discussion is very clearly written. At the end the authors commented that cell lines do not fully represent heterogeneity in tumor tissue, which is a limitation of this study and that they are planning in the future to test the clinically relevant p38 MAPK inhibitor ralimetinib in patient derived organoids. I suggest to authors to extend the discussion in terms of the possible consequences of p38 inhibition on tumor microenvironment in advanced stage, knowing the importance of intercellular communication as well as metabolic specificities of constituents and their interplay. pEMT cells possess character of stemness and mostly underwent to metabolic shift to a glycolytic mode of energy production even in the presence of oxigen. It would be very beneficial to predict some of the consequences that abolished activity of these cells through p38 suppression might have on TME and subsequently, whole process of disease progression and therapy response.
The authors are grateful for the comments. We discuss the anti-inflammatory potential of p38 inhibitor ralimetinib on the TME and Slug stabilization, which could be an important point of our further work. Page 16, Lines 582-606.
Minor corrections:
126 and 127. Please rephrase.
The required text revision is done. Page 4, Lines 134-137.
141/142. Please rephrase. The term of biological repeats is unclear. It is enough to tell that all exp. were repeated tree times
Corrected as suggested. Page 4, Line 149.
144-145 incomplete
We now added further information and completed the sentence. Page 4, Lines 151-152.
145- Rephrase the title
Corrected as suggested. Page 4, Line 150.
199- Title should be changed. Maybe like: Flowcytometric assessment of……
Corrected as suggested. Page 4, Line 206.
240-242. I suggest to write this as a the separate paragraph in Material and methods section, since this is microscopically evaluation, and it is not a part of flowcytometry, even the authors use the cells prepared for it to visually explore them.
This text passage is now part of the methods section. Page 5, Lines 225-227.
Round 2
Reviewer 1 Report
Comments and Suggestions for Authors
The authors answered my questions. The paper can now be accepted for publication